# Physicochemical Parameters, Phytochemical Profile and Antioxidant Properties of a New Beverage Formulated with Xique-Xique (*Pilosocereus gounellei*) Cladode Juice

**DOI:** 10.3390/foods10091970

**Published:** 2021-08-24

**Authors:** Julia M. C. de Oliveira, Evandro L. de Souza, Kaíque Y. G. de Lima, Marcos dos S. Lima, Vanessa B. Viera, Rita de Cássia R. do E. Queiroga, Maria Elieidy G. de Oliveira

**Affiliations:** 1Department of Nutrition, Health Sciences Center, Federal University of Paraíba, João Pessoa 58051-900, Brazil; julia.caju@academico.ufpb.br (J.M.C.d.O.); els@academico.ufpb.br (E.L.d.S.); rcqueiroga@academico.ufpb.br (R.d.C.R.d.E.Q.); 2Department of Biotechnology, Biotechnology Center, Federal University of Paraíba, João Pessoa 58051-900, Brazil; kaique.gervazio@academico.ufpb.br; 3Department of Food Technology, Federal Institute of Sertão Pernambucano, Petrolina 56302-100, Brazil; marcos.santos@ifsertao-pe.edu.br; 4Department of Nutrition, Center of Education and Health, Federal University of Campina Grande, Cuité 58175-000, Brazil; vanessa.bordin@professor.ufcg.edu.br

**Keywords:** cactus, functional beverage, physicochemical characteristics, bioactive compounds, antioxidant activity

## Abstract

This study elaborated different formulations with xique-xique (*Pilosocereus gounellei*) cladode, passion fruit and lime juice and sugar cane syrup. The formulated beverages were subjected to physical and physicochemical analysis, determination of total carotenoid, total flavonoid and total phenolic compound contents, as well as of their antioxidant activity (ABTS and FRAP method), organic acid, sugar and phenolic compound profile during 21 days of refrigeration storage (4 °C). Significant variations were found among formulated beverages for most of the measured parameters during storage. Beverages with xique-xique juice were less acidic (7.90–10.27 g/100 mL) than beverages without this juice (11.66–12.76 g/100 mL). Beverages with xique-xique juice had overall higher contents of bioactive compounds and higher antioxidant activity when compared to the control formulation. Beverages with the highest xique-xique juice concentrations had the highest contents of carotenoids (51.51–59.27 µg/100 mL), flavonoids (1.39–2.15 mg CE/100 mL), phenolic compounds (68.49–115.66 mg EGA/100 mL) and antioxidant activity, as measured by ABTS (0.71–0.84 µmol Trolox/mL) and FRAP (0.33–0.39 µmol Trolox/mL). These results indicate that the incorporation of xique-xique cladode juice in these mixed beverages enhanced their bioactive properties, especially of antioxidant compounds, enabling the development of a new product with potential functional properties to the beverage industry.

## 1. Introduction

The search for functional products and their diffusion in the food market have increased in the last years mainly due to the growing incidence of chronic non-communicable diseases and the raising awareness of consumers about the relationship between diet and health, leading to a high demand for healthier foods [1,2]. In this context, the formulation of beverages with natural ingredients with the capability of exerting specific health-related physiological functions has been a major focus of interest for the food industry [3].

The world market for functional beverages is expected to grow around 7.8% by the year of 2022, being the fastest growing category among the functional foods [4]. This market growth could be associated with the easier transport and storage of the functional beverages, as well as with the greater possibility of incorporating components and bioactive nutrients in their formulation, which add desired functionality to these products [3,5].

Vitamins, minerals, amino acids, fatty acids, prebiotic fibers, probiotic bacteria and antioxidant compounds are among the most common components used for the formulation of functional beverages [6]. Some studies have focused on the investigation of unconventional matrices as new sources of bioactive components, such as exotic fruits, succulent plants and cacti, envisaging their use to the development of new functional beverages [7,8,9].

Xique-xique (*Pilosocereus gounellei* A. Weber ex K. Schum. Bly. ex Rowl), Cactaceae family, is an endemic plant species from the Caatinga biome in Northeastern Brazil, being still an underexplored non-conventional plant food [10]. This species is widely used by a population from this region in folk medicine, its roots, flowers, fruits and cladodes commonly being used for the treatment of urethra and prostate inflammation [11], constipation [12], gastritis [13] and jaundice [14]. Xique-xique has been also used for human consumption, its fruits being consumed in natura, while the cladodes are used for the production of candies, flour, bakery products and juices [12,15,16].

Early studies have found a variety of bioactive compounds in xique-xique cladodes, such as phenolic compounds, flavonoids and betalains [10,17]. The contents of these bioactive compounds in xique-xique, especially of phenolic compounds, have been shown to correlate with its high antioxidant capacity. Soluble and insoluble fibers have been also found in xique-xique cladode juice [18,19].

Results of in vivo studies have shown that saline extract from stem, roots and xique-xique cladodes exert antinociceptive, antipyretic, hypoglycaemic and hypolipemic effects in rats with no evidence of toxic effects [20,21]. The consumption of xique-xique cladode juice was reported to have protective effects toward intestinal inflammation and decrease oxidative stress in rats with inflammatory bowel disease, besides having no evident toxic effects [19]. The results of these studies indicate that xique-xique could be safe for human consumption and a promising raw material to formulate potentially functional foods and beverages.

Considering that xique-xique cladode pulp does not have a pronounced flavor, the addition of ingredients to add flavor in xique-xique-based beverage formulations could allow an improvement in their sensory acceptance. Lime (*Citrus latifolia* Tanaka) and passion fruit (*Passiflora edulis* Sims. f. flavicarpa Deg.) are popular fruits widely used for beverage formulations and well-accepted by consumers, besides being sources of vitamins, minerals and antioxidant compounds [22,23]. It has been shown that the addition of fruit blends in beverage formulations could enhance the nutritional and sensory characteristics when compared to their addition separately [24]. The addition of sugar cane syrup could also contribute to the improvement of sensory aspects while also adding nutritional and functional value to the beverage, considering its high mineral contents, mainly of potassium, calcium and magnesium [25].

Considering the potential of xique-xique cladodes in regards to the variety of bioactive compounds found in their composition, the formulation of beverages using xique-xique cladode juice in combination with passion fruit and lime juice and sweetened with sugar cane syrup would enable innovation for the use of this still underexplored plant species. This study aimed to formulate potentially functional beverages with xique-xique, passion fruit and lime juice and cane syrup, as well as to evaluate its physicochemical parameters, content of total carotenoids, flavonoids and phenolic compounds, profile of phenolic compounds and antioxidant activity during refrigeration storage.

## 2. Materials and Methods

### 2.1. Raw Materials and Ingredients

Xique-xique (*Pilosocereus gounellei*) cladodes were collected from a private cultivation area located at the municipality of Boa Vista (coordinates 07°15′3 ″ S 36°14′24″ O, Paraíba, Brazil). The plant was identified by Prof. Dr. Leonardo Person Felix (Center of Agricultural Sciences, Federal University of Paraíba, Bananeiras, Paraíba, Brazil), and a certified voucher specimen (number 15437) was deposited in Herbarium Prof. Jaime Coelho Morais (Center of Agricultural Sciences, Federal University of Paraíba, Bananeiras, Paraíba, Brazil). Passion fruit (*Passiflora edulis* Sims. f. flavicarpa Deg.) and tahiti lime (*Citrus latifolia* Tanaka) at commercial maturation stage and sugar cane syrup (Gascana Doces e Rapaduras, Natal, Rio Grande do Norte, Brazil) were purchased from local supermarkets (João Pessoa, Paraíba, Brazil).

### 2.2. Preparation of Xique-Xique, Passion Fruit and Lime Juice

The xique-xique cladodes had their spikes removed with a domestic knife and were cleaned with running water and sanitized by immersion in chlorinated water (100 ppm, 15 min) and peeled. To prepare the xique-xique cladode juice, the central stems of cladodes were removed and discarded. Cladode pulp was collected and mashed with a domestic blender (Mondial Turbo L-900 FR, Barueri, São Paulo, Brazil) without adding water. Mashed pulp was firstly filtered with a sieve (20 mesh) and then with a 100% sterile muslin cloth. Filtered juice was subjected to heating (90 ± 1 °C, 5 min), cooled in an ice bath (10 ± 1 °C), stored in polyethylene terephthalate (PET) bottles and kept under refrigeration (4 ± 1 °C). 

The passion fruit juice was obtained by mashing the fruit pulp with a domestic blender (Mondial Turbo L-900 FR) and filtering through a sieve (20 mesh) with potable water in a ratio of 1:4. Lime juice was obtained with a domestic juicer and potable water added in the proportion of 1:1. Prepared juices were stored in PET bottles under refrigeration (4 ± 1 °C). Obtained xique-xique cladode, passion fruit and lime juices were used to preparate the mixed beverages within a maximum period of 12 h of refrigeration storage.

### 2.3. Formulation of Beverages

The beverages were formulated with mixtures of xique-xique cladode pulp juice, passion fruit juice, lime juice and cane syrup in different rates, as shown in Table 1. Four beverage formulations were prepared, one being the control beverage (without xique-xique cladode juice) and three formulations with 30%, 40% and 50% xique-xique cladode juice (*v*/*v*). After mixing and homogenizing the ingredients, the beverages were packed in PET bottles and stored under refrigeration (4 ± 1 °C) up to the characterization analysis. The beverages were evaluated on days 1, 7, 14 and 21 of refrigeration storage.

### 2.4. Physical and Physicochemical Characterization of Formulated Beverages 

The beverages were characterized regarding their physical and physicochemical parameters according to standard procedures [26], to cite: determination of molar acidity by titration; electrometric determination of pH using a potentiometer with combined glass electrode (Model Q400AS, São Paulo, Brazil); total soluble solids (TSS) (°Brix, g/100 mL) using a portable refractometer (HI96801, Hanna instruments, São Paulo, Brazil) at 25 ± 1 °C; ash content determined by carbonization and incineration in a muffle furnace stabilized at 550 °C; quantification of protein content by the Kjedahl method with a conversion factor of 6.25 multiplied by the percentage of nitrogen and total sugars according to the Fehling reduction methodology. Total lipid content was measured using the method of Folch, Lees, and Stanley, with modifications [27]. Briefly, lipids were extracted with chloroform/methanol (2:1, *v*/*v*) in a sample to solvent ratio of 1:15, and homogenized with a mini-Turrax apparatus (TE-102, Tecnal, Piracicaba, São Paulo, Brazil) for 2 min. The samples were filtered, added with 1.5% Na_2_SO_4_ (20%, *v*/*v*), mixed and allowed to stand until it separated into two phases, where the lower phase was recovered and the solvents evaporated using a drying oven at 90 °C. The samples were kept in a desiccator to reach room temperature, and the lipid content was weighted. 

### 2.5. Determination of Sugar and Organic Acid Profile of Formulated Beverages

For the extract preparation, a 10 g-aliquot of each beverage was homogenized with a mini-Turrax apparatus (Tecnal), centrifuged (9000× *g*, 15 min, 4 °C) and filtered with a 0.45 µm-filter (Millex Millipore, Barueri, São Paulo, Brazil). Contents of sugars and organic acids were simultaneously measured with the HPLC—diode array detector (DAD) (G1315D model)—refractive index detector (RID) (G1362A model) using an Agilent chromatograph (model 1260 Infinity LC, Agilent Technologies, St. Clara, CA, USA) equipped with a quaternary solvent pump (G1311C model), degasser, thermostatic column compartment (G1316A model) and automatic auto-sampler (G1329B model) according to a previously described method for simultaneous sugar and organic acid detection [28]. The other analytical conditions were: an Agilent Hi-Plex H column (8 μm, 7.7 × 300 mm); mobile phase H_2_SO_4_ 4 mM/L in ultrapure water; flow rate of 0.7 mL/min; separation temperature of 70 °C and sample injection volume of 10 μL. 

Organic acids were detected by DAD at 210 nm, and sugars were detected by RID. Data were processed with OpenLAB CDS ChemStation EditionTM software (Agilent Technologies). HPLC sample peaks were identified by a comparison of their retention times with those of organic acid and sugar standards. Mean peak areas were used for the quantification. Glucose and fructose standards were obtained from Sigma-Aldrich (St. Louis, MA, USA), and citric, malic, succinic and lactic acid standards were obtained from Vetec Química Fina (Rio de Janeiro, Brazil). Organic acids and sugar contents were expressed as g per 100 mL of the formulated beverage (g/100 mL).

### 2.6. Determination of Total Carotenoid Content in Formulated Beverages

Total carotenoid content was measured according to a previously described method [29]. Absorption was measured at 470, 645 and 662 nm with a spectrophotometer (Bel Photonics, Piracicaba, São Paulo, Brazil). Total carotenoid content was expressed as µg per 100 mL of the formulated beverage (µg/100 mL).

### 2.7. Total Flavonoid and Total Phenolic Compound Contents in Formulated Beverages

For the extract preparation, a 2 g-aliquot of each beverage formulation was homogenized with 20 mL of 80% methanol for 10 min using a mini-Turrax apparatus (Tecnal), kept to rest for 24 h and filtered with a 125 mm-filter paper (Whatman, GE Healthcare, Chicago, IL, USA). Total flavonoid content was measured according to a previously described procedure, with modifications [30]. A 0.5 mL-aliquot of the extract was added to 2 mL of distilled water and homogenized with 150 µL of a 5% sodium nitrite. After 5 min, 150 µL of 10% aluminum chloride solution was added and, after 6 min, 1 mL of 1 M sodium hydroxide solution and 1.2 mL of distilled water were added to the mixture. Sample absorbance was measured at 510 nm with a spectrophotometer (BEL Photonics) against a blank in the absence of extract. The content of total flavonoids was determined with a standard curve of catechin (Sigma-Aldrich) equivalents (CE). Results were expressed as mg catechin equivalents (CE) per 100 mL of the sample (mg CE/100 mL).

Total phenolic content was determined with the Folin-Ciocalteu method [31]. A 250 µL-aliquot of the extract was homogenized with 1250 µL of 10% Folin-Ciocalteau reagent, stirred with a Vortex mixer (Quimis, Diadema, São Paulo, Brazil), kept at room temperature (25 ± 0.5 °C) under the dark for 6 min, added with a 1 mL-aliquot of 7.5% sodium carbonate solution and placed in a water-bath (Raypa, Barcelona, Spain) at 50 ± 0.5 °C for 5 min. Absorbance was measured at 765 nm with a spectrophotometer (Bel Photonics). A blank was performed with the absence of extract to reset the spectrophotometer. Total phenolic content was determined with a standard curve prepared with gallic acid (Sigma-Aldrich). Results were expressed as mg equivalent of gallic acid (EGA) per 100 mL of the sample (mg EGA/100 mL).

### 2.8. Determination of Phenolic Compound Profile in Formulated Beverages

For the extract preparation, a 5 g-aliquot of each beverage was homogenized with 5 mL of 80% methanol (Sigma Aldrich) using a mini-Turrax apparatus (Tecnal), centrifuged (9000× *g*, 15 min, 4 °C) and filtered with a 0.45 µm-filter (Millex Millipore). Individual phenolic compounds were determined with a high-performance liquid chromatograph, with gradient adaptations and a runtime to quantify different phenolic classes, using an Agilent 1260 Infinity System LC liquid chromatograph (Agilent Technologies) coupled to a diode array detector (DAD) (G1315D). The column was a Zorbax Eclipse Plus RP-C18 (100 × 4.6 mm, 3.5 μm), and the pre-column was a Zorbax C18 (12.6 × 4.6 mm, 5 μm) (Agilent Technologies). The oven temperature was 35 °C, and the injection volume was 20 μL diluted in phase A and filtered with a 0.45 μm-filter (Millex Millipore). 

Solvent flow was 0.8 mL/min; a new gradient used in separation was 0 to 5 min: 5% B; 5 to 14 min: 23% B; 14 to 30 min: 50% B; 30–33 min: 80% B, where solvent A was a solution of phosphoric acid (0.1 M, pH = 2.0), and solvent B was acidified methanol with 0.5% H3PO4. Data were processed with OpenLAB CDS ChemStation Edition software (Agilent Technologies). Detection of phenolic compounds was conducted at 220, 280, 320, 360 and 520 nm. Identification and quantification were conducted by a comparison with external standards (Sigma-Aldrich). Results were expressed as mg of phenolic compound for 100 mL of the beverage (mg/100 mL) [32].

### 2.9. Evaluation of Antioxidant Activities of Formulated Beverages

The FRAP and ABTS methods were used to evaluate the antioxidant activity of formulated beverages. Extracts were prepared as described in Section 2.7. The capability of these extracts of reducing iron was measured with the FRAP (Ferric Reducing Ability of Plasma) method according to a previously described procedure, with modifications [33]. The FRAP reagent was prepared with 3 mol/L of acetate buffer (pH 3.6) + 10 mM/L of TPTZ (2,4,6-tris (2-pyridyl)-s-triazine) in a 40 mM/L HCl solution + 20 mM FeCl_3_. A 200 mL-aliquot of the extract was added to 1800 µL of the FRAP solution, stirred with a Vortex mixer (Quimis) for 30 s and placed in a water bath for 30 min at 37 °C. Absorbance was measured at 593 nm with a spectrophotometer (Bel Photonics). The standard curve was created with Trolox 1 mM, and results were expressed in μmol Trolox per mL of the sample (µmol Trolox/mL).

The extract ability to capture ABTS^•+^ cation (2,2-azino-bis (3-ethylbenzothiazoline)-6-sulfonic acid) was measured with the ABTS method. The ABTS reagent, which was prepared by mixing 5 mL of 7 mM ABTS with 88 µL of 140 mM potassium persulfate (final concentration of 2.45 mM), and ABTS^•+^ was formed after resting the ABTS reagent for 12–16 h at room temperature (25 ± 0.5 °C) under the dark [34]. The ABTS^•+^ solution was diluted with distilled water to an absorbance value of 0.800–0.900 at 734 nm. Absorbance of the reaction mixture (600 µL) with 100 µL of extract and 500 µL of ABTS^•+^ solution was measured at 734 nm in a spectrophotometer (Bel Photonics). A control solution with 100 µL of extracting solvent + 500 µL of ABTS radical was prepared. The negative control solution was the extracting solvent for each extract used to reset the spectrophotometer. Trolox was used as a reference, and results were expressed in µmol Trolox equivalent antioxidant capacity per mL of the sample (µmol Trolox/mL).

### 2.10. Statistical Analysis

Experiments were conducted in triplicate in three independent repetitions, and results were expressed as the average ± standard deviation. Data were submitted to analysis of variance (ANOVA) followed by Tukey’s test using *p* < 0.05. Correlations were calculated with Pearson’s correlation coefficient (r). A principal component analysis (PCA) was conducted to evaluate the correlation matrix among total carotenoids, total flavonoids, total phenolic compounds and antioxidant activity during refrigeration storage. Statistical analysis was conducted with computational software GraphPad Prism 9.0 (GraphPad Software Inc., San Diego, CA, USA).

## 3. Results and Discussion

### 3.1. Physical and Physicochemical Characteristics of Beverages

The results of the physical and physicochemical parameters of the formulated beverages during 21 days of refrigeration storage are shown in Table 2. There was a reduction in pH values and an increase in tritatable acidity (TA) during storage (*p* < 0.05) in most beverage formulations. The control formulation (CB) had higher acidity when compared to other beverage formulations (*p* < 0.05). This variation could be attributed to the higher concentration of passion fruit juice in CB, since it is a more acidic juice with a pH of approximately 2.8, while xique-xique cladode juice has a pH of approximately 5.0, as reported by previous studies [18,35]. 

The increasing acidity found during the measured refrigeration storage period could be caused by degradation of polyphenols in beverage formulations over time [9]. This gradual increase in acidity could be a positive feature in formulated beverages, as a low pH could inhibit the growth of pathogenic and spoilage microorganisms [9].

An increase (*p* < 0.05) in TSS values from day 1 to day 21 of storage was found in all formulations. This increase in TSS values at the later storage period might be due to the hydrolysis of polysaccharides into monosaccharides and oligosaccharides in beverages, as this parameter is an indicator of sugar contents, primarily sucrose, glucose and fructose. Similar results have been found in earlier studies with functional beverages with *Aloe vera* [9] and cucumber-melon [5].

The CB formulation had higher TSS contents than other beverage formulations during the measured storage period (*p* < 0.05). This variation could be attributed to different sugar concentrations in ingredients used to elaborate the beverages, as passion fruit juice has a higher TSS value (13.17 °Brix) than xique-xique cladode juice (5.45 °Brix) [15,36].

Total ash values were higher (*p* < 0.05) in B40 and B50 formulations (maximum values of 0.40 and 0.47 g/100 mL, respectively) when compared to the CB formulation (maximum value of 0.26 g/100 mL). High ash contents have been found in xique-xique cladodes and derived juice [15,18], which should be linked to high contents of minerals in these products. A previous study found high contents of potassium, magnesium and calcium in xique-xique cladode juice [37]. Therefore, it could indicate that the addition of xique-xique cladode juice to the beverage formulations contributed to the higher mineral contents found in these beverages.

Similar to TSS, the total sugar contents had an increase (*p* < 0.05) from day 1 to day 21 of storage for the examined beverage formulations, with CB having overall a higher TSS content when compared to formulations with xique-xique cladode juice. This increase could be attributed to the hydrolysis of complex sugars into simple sugars due to a higher acidity, as previously found for a functional *Aloe vera*-based beverage [9]. Overall, the protein and lipid contents did not change during the measured storage period or among the different beverage formulations (*p* ≥ 0.05).

### 3.2. Soluble Sugar and Organic Acid Contents of Beverages

Glucose and fructose were the sugars found in the formulated beverages, glucose being found in the highest contents (Table 3). CB and B50 formulations had the overall highest contents of glucose (1.05 ± 0.01 and 1.09 ± 0.03 g/100 mL, respectively) and fructose (1.04 ± 0.15 and 1.03 ± 0.04 g/100 mL, respectively). Only B40 showed a significant increase (*p* < 0.05) from day 1 to day 21 of storage for glucose (0.55 ± 0.01 to 1.02 ± 0.04 g/100 mL) and fructose (0.67 ± 0.02 to 0.98 ± 0.04 g/100 mL) contents.

Previous studies have also found glucose and fructose in xique-xique cladode juice (0.28 ± 0.05 g/100 g and 0.21 ± 0.06 g/100 g, respectively) [18] and passion fruit juice (1.41 ± 0.01 and 1.46 ± 0.01 g/100 g, respectively) [38], which have shown that xique-xique cladode juice has a slightly lower sugar content than passion fruit juice. Glucose and fructose contents found in the formulated beverages could be attributed to the combination of xique-xique cladode and passion fruit juices, as well as to other ingredients used in their preparation, especially sugar cane syrup.

Organic acids play an important role in taste, flavor and consumer acceptance of fruit beverages. Citric, succinic, malic and lactic acids were the organic acids found in the examined beverage formulations (Table 3). Citric acid was the major organic acid regardless of the examined beverage formulation (0.72–1.12 g/100 mL). Overall, the contents of citric acid decreased as the concentration of xique-xique juice increased in the beverage formulations, indicating that passion fruit juice should be the main source of citric acid in these products. 

Regarding the lactic acid contents, a significant difference was found among the beverage formulations on day 21 of storage, where B40 and B50 had the highest lactic acid contents (0.03 ± 0.00 g/100 mL). Lactic acid is mostly produced by lactic acid bacteria (LAB), and its production occurs naturally in a variety of dairy, meat and plant food products, with an important role as a natural biopreservative due to its well-known antimicrobial effects [39]. No difference was found (*p* ≥ 0.05) for malic and succinic acid contents among the examined beverage formulations. Overall, the contents of organic acids in the formulated beverages did not vary during the measured storage time (*p* ≥ 0.05).

### 3.3. Total Contents of Carotenoids, Flavonoids and Phenolic Compounds of Beverages 

Among the examined beverage formulations, B50 had the highest contents of total phenolic compounds during storage (115.66 ± 0.10 to 68.49 ± 0.01 mg EGA/100 mL), while CB had the lowest contents (59.06 ± 0.10 to 40.19 ± 0.00 mg EGA/100 mL). Therefore, the higher the concentration of xique-xique cladode juice in the beverage formulations, the higher was the total phenolic content (*p* < 0.05) (Figure 1A). Phenolic compounds are originated from a class of secondary metabolites in plants, with a variety of bioactivities beneficial to human health, outstanding their antioxidant activities [40]. Different phenolic compounds have been found in xique-xique cladode juice, which have been linked to the anti-inflammatory and antioxidant effects in acetic acid-induced colitis in animal models [19], reinforcing the great functional potential of this product.

A decrease (*p* < 0.05) in total phenolic compound contents was found during storage in all examined beverage formulations (Figure 1A). A decrease in total phenolic compounds was also found in pasteurized orange passion during refrigeration storage [36]. This reduction in total phenolic compound contents in the formulated beverages, even under low temperature storage, could be related to the high susceptibility of these compounds to chemical reactions and enzymatic oxidation [41].

The B50 formulation had the highest content of total flavonoids on day 1 of storage (2.15 ± 0.00 mg EC/100 mL); however, this formulation had a lower content of total flavonoids (*p* < 0.05) on day 21 of storage when compared to other beverage formulations (1.39 ± 0.01 mg CE/100 mL), although it still had a considerable flavonoid concentration (Figure 1B). Flavonoids are a class of phenolic compounds with antioxidant, anti-inflammatory and antimicrobial activities [10,40]. Early studies had also found flavonoids in xique-xique cladode ethanol extract [10] and juice [19], as well as in other *Pilosocereus* species [42]. A decrease in flavonoid contents during refrigeration storage was also found in a mixed beverage with hibiscus and coconut water [43].

Similar to the total phenolic compounds, carotenoid content increased as the concentration of xique-xique cladode juice used in the beverage formulation increased, where B50 had the highest total carotenoid contents (59.27 ± 0.00 to 51.51 ± 0.00 µg/100 mL). Carotenoids, also known as provitamin A, play an important role in antioxidant and immune system activity. The consumption of carotenoid-rich foods is an important dietary aspect to be controlled, since it is a nutrient not synthesized by the human body [44].

There is no data in available literature regarding the determination of carotenoids in xique-xique cladodes or derived products. Total carotenoid contents decreased (*p* < 0.05) during storage regardless of the examined beverage formulation (Figure 1C). Losses in carotenoid contents in fruit-based beverages have been shown to depend usually on the fruit maturity stage, as well as on the storage and processing conditions, the susceptibility to the oxidation being the main cause for these losses [45]. 

### 3.4. Phenolic Compound Profile of Beverages

The phenolic compound profile identified in the formulated beverages is shown in Table 4. Flavanones, flavonols, flavanols, anthocyanins hydroxybenzoic acids, hydroxynamic acids and polyphenols were found in the beverage formulations during refrigeration storage, totaling 18 phenolic compounds. Flavanones were found at the lowest contents, and only CB and B30 had hesperidin in their composition. Naringenin was found in all beverage formulations on day 1 of storage, but this compound was found only in B30 up to day 14 of storage.

Myricetin, epigallocatechin gallate, epicatechin gallate, kaempferol and procyanidin A2 were the most prevalent phenolic compounds in the examined beverage formulations. Myricetin was the major phenolic compound found in the beverage formulations, and its contents did not differ among them (*p* ≥ 0.05). CB had higher contents of syringic acid, epicatechin gallate and epicatechins when compared to B50, indicating that these compounds could be mostly attributed to the passion fruit juice. Presence of epicatechin and syringic acid was previously found in *p. edulis* fruit pulp [46], which is in agreement with the results of this study.

The B50 formulation had a higher procyanidin A2 content (0.15 ± 0.00 to 0.13 ± 0.00 mg/100 g) when compared to CB (0.13 ± 0.00 to 0.12 ± 0.01 mg/100 g), indicating that this compound could be found in higher contents in xique-xique cladode juice. Procyanidins are phytoallexins found in vegetables and fruits have been related to antioxidant, anticancer, antibacterial, anti-inflammatory, cardioprotective and immunomodulatory effects [47]. 

In general, the contents of phenolic compounds found in the examined beverage formulations had few variations during storage. Still, most of the phenolic compounds did not have significant content variations among the examined beverage formulations. These results could be in disagreement with the results of total phenolic contents determined with the Folin-Ciocalteu reagent (FCR) method, where the examined beverage formulations, especially B50, had higher total phenolic contents when compared to CB. An early study also found higher contents of phenolic compounds when determined by the FCR method than those when determined by the HPLC technique, it being suggested that such a difference could be due to the reaction of the Folin-Ciocalteu reagent not only with phenolic compounds, which could affect the results of this determination [7]. However, it should be also considered that, in the present study, the chromatograms had peaks not identified due to a limitation in the number of available external standards, which may have undervalued the contents of phenolic compounds determined by the HPLC technique in the examined beverage formulations. 

### 3.5. Antioxidant Activity of Beverages 

Regarding the antioxidant activity determined with either the FRAP or ABTS method, the beverage formulations with xique-xique cladode juice had higher antioxidant activity (*p* < 0.05) when compared to CB (Figure 2). When determined by the FRAP method, the formulations B50 and B40 had the highest antioxidant activities (*p* < 0.05), both measuring 0.394 ± 0.00 µmol Trolox/mL on day 1 of storage and 0.331 and 0.336 ± 0.00 µmol Trolox/mL, respectively, on day 21 of storage. CB had values ranging from 0.344 to 0.297 ± 0.00 µmol Trolox/mL during storage. Similar results were found with the ABTS assay, where B50 had the higher radical scavenging activity on days 1 and 7 of storage (0.840 and 0.810 ± 0.00 µmol Trolox/mL, respectively), while B40 had the higher radical scavenging activity on days 14 and 21 of storage (0.810 and 0.770 ± 0.00 µmol Trolox/mL, respectively). CB had lower antioxidant activities ranging from 0.680 and 0.620 ± 0.00 µmol Trolox/mL (*p* < 0.05). 

These results are in agreement with an early study that found free-radical scavenging activity in ethanol extracts of xique-xique, where extracts from its fruits and cladodes had the highest activities in both FRAP and ABTS assays [10]. The antioxidant activity decreased (*p* < 0.05) in examined beverage formulations during storage when measured by the FRAP and ABTS method. The decrease in antioxidant activity during refrigeration storage should be expected, since this activity could be mostly attributed to the presence of bioactive components (e.g., vitamins and phenolic compounds), which are usually thermosensitive and susceptible to oxidation [7].

### 3.6. Correlation between Antioxidant Activity and Bioactive Compounds in Beverages

The correlation between antioxidant activity (ABTS and FRAP), bioactive compounds (carotenoids, flavonoids and phenolic compounds) and the four formulate beverages during refrigeration storage is shown in Figure 3. Data variance was explained by 99.90% for PC1 and 0.07% for PC2. The strongest negative correlation with PC2 was found for total flavonoids, while the weakest positive correlation was found for total carotenoids. ABTS had a strong positive correlation with PC2. Total phenolic compounds had a strong positive correlation with PC1, while FRAP had a very strong positive correlation with PC1 and PC2. 

Possibly, CB formulation (shown as yellow) moved away from total phenolic contents and total carotenoids, which may confirm the results shown in Figure 1 where this formulation had the lowest content for these compounds. Beverages formulated with xique-xique cladode juice (B30, B40 and B50) are closer to total phenolic and carotenoid contents and antioxidant activity assays. As the storage time passed, these formulations moved away from total phenolic and total flavonoid contents, but they stayed in relation to total carotenoid contents, which were the bioactive compounds less affected by storage time.

Regarding the correlations among antioxidant assays and bioactive compounds (Figure 3, for the ABTS method), the strongest positive correlation was found for total carotenoids (r = 0.81), followed by total phenolic compounds (r = 0.80) and total flavonoids (r = 0.68). As for FRAP, total phenolic compounds and total flavonoids were strongly positively correlated (r = 0.85), while total carotenoids had a lower correlation (r = 0.66). Therefore, these correlations show that antioxidant activity measured by the ABTS method was more affected by total carotenoid and total phenolic contents, while antioxidant activity measured by FRAP was more affected by flavonoid and total phenolic contents. These results could explain the higher antioxidant activities measured with ABTS and FRAP found for the formulations with xique-xique cladode juice, mainly B40 and B50, which had the best results for total phenolic compounds, total flavonoids and total carotenoids among the formulated beverages.

## 4. Conclusions

The results of this study indicated significant variations in the measured physicochemical and phytochemical parameters among the four beverages formulated through refrigeration storage time, where beverages with xique-xique cladode juice had lower acidity, and total soluble solids and total sugar contents, when compared to CB. Beverages formulated with xique-xique cladode juice had overall an increased content of total flavonoids, total carotenoids and total phenolic compounds, as well as a higher antioxidant activity when compared to CB during 21 days of refrigeration storage. The beverage formulation with the highest concentration of xique-xique cladode juice had the best results regarding the contents of total ash, total flavonoids, total carotenoids and total phenolic compounds and antioxidant activity, indicating its potential as a new functional mixed beverage. Further investigations, such as in vivo analysis, could improve the development of the proposed xique-xique cladode mixed beverages and their insertion in the food market as a potentially functional product with health-related properties.

## Figures and Tables

**Figure 1 foods-10-01970-f001:**
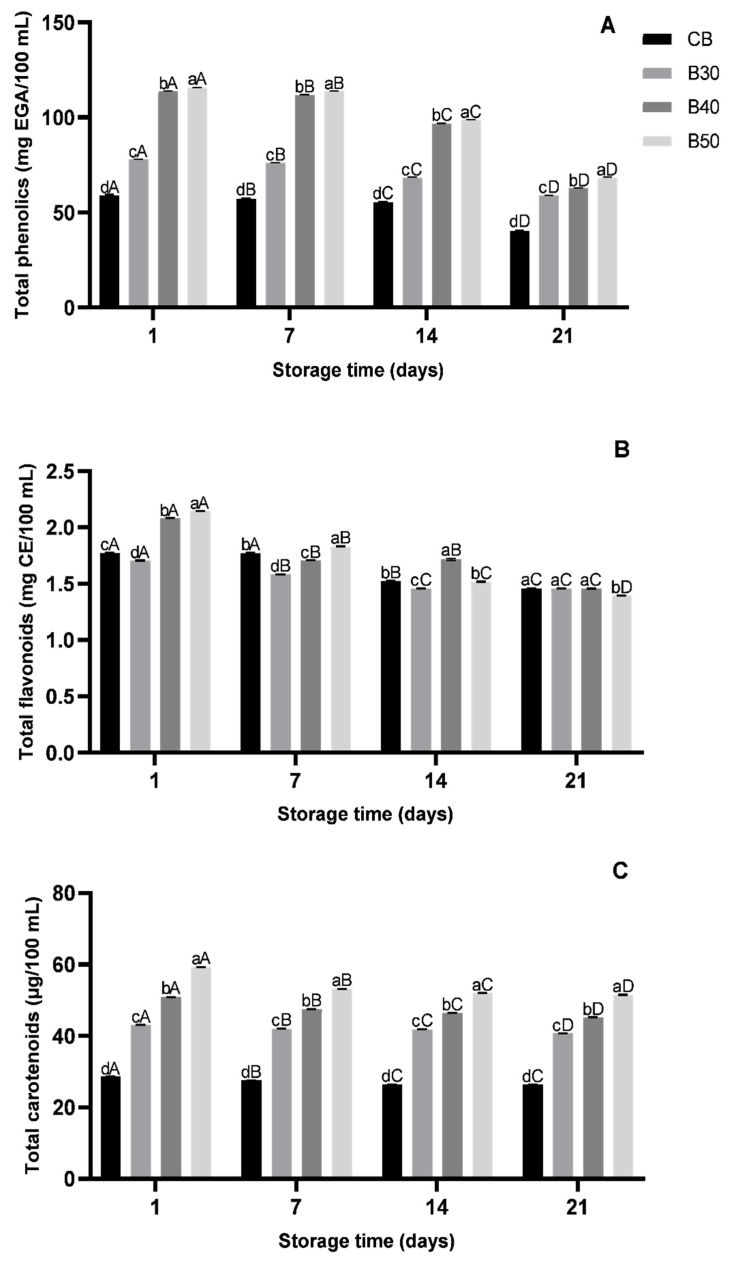
Contents of bioactive compounds in formulated beverages during 21 days of refrigeration storage (4 °C). (**A**) Total phenolic compounds, (**B**) total flavonoid and (**C**) total carotenoid contents in four formulated beverages. Error bars represent the standard deviation of the mean (*n* = 3). Different lowercase letters on the same storage time differ (*p* < 0.05) among beverage formulations, and different uppercase letters for the same beverage formulation differ (*p* < 0.05) among different storage time periods based on Tukey’s test. CB—control beverage; B30—beverage with 30% of xique-xique cladode juice; B40—beverage with 40% of xique-xique cladode juice; B50—beverage with 50% of xique-xique cladode juice.

**Figure 2 foods-10-01970-f002:**
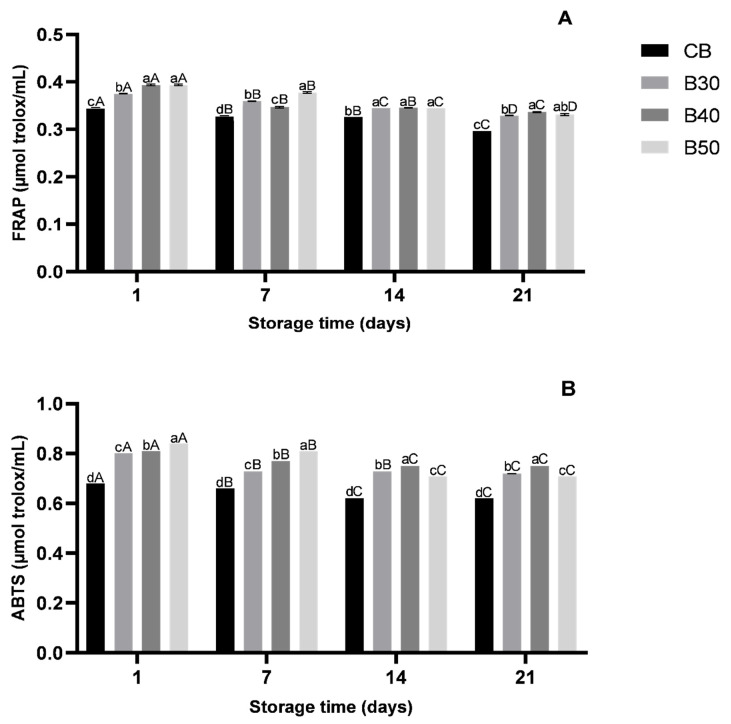
Antioxidant activity in formulated beverages during 21 days of refrigeration storage (4 °C) by (**A**) ferric reducing antioxidant power (FRAP, µmol Trolox/mL) and (**B**) ABTS radical cation scavenging activity (µmol Trolox equivalent/mL). Error bars represent the standard deviation of the mean (*n* = 3). Different lowercase letters on the same storage time period differ (*p* < 0.05) among formulated beverages, and different uppercase letters on the same beverage formulation differ (*p* < 0.05) among different storage time periods based on Tukey’s test. CB—control beverage; B30—beverage with 30% of xique-xique cladode juice; B40—beverage with 40% of xique-xique cladode juice; B50—beverage with 50% of xique-xique cladode juice.

**Figure 3 foods-10-01970-f003:**
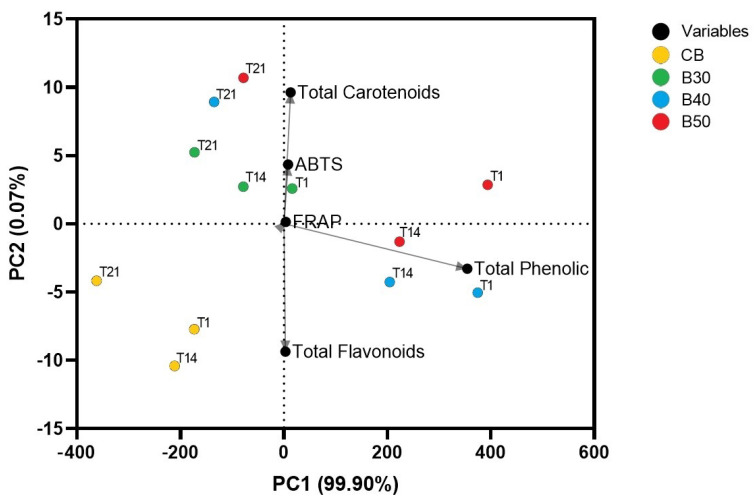
Principal Component Analysis of contents of total carotenoids, total flavonoids and total phenolic compounds and antioxidant activity (FRAP and ABTS) of formulated beverages (*n* = 3) on days 1, 14 and 21 of refrigeration storage (T1, T14 and T21, respectively) (4 °C). CB—control beverage; B30—beverage with 30% of xique-xique cladode juice; B40—beverage with 40% of xique-xique cladode juice; B50—beverage with 50% of xique-xique cladode juice.

**Table 1 foods-10-01970-t001:** Formulation of beverages with xique-xique cladode, passion fruit and lime juices and sugar cane syrup.

Ingredients	Formulations ^1^			
	CB	B30	B40	B50
Xique-xique juice	-	30%	40%	50%
Passion fruit juice	80%	50%	40%	30%
Lime juice	15%	15%	15%	15%
Sugar cane syrup	5%	5%	5%	5%

^1^ Values presented in percentage in relation to beverages total volume. CB—control beverage; B30—beverage with 30% of xique-xique juice; B40—beverage with 40% of xique-xique juice and B50—beverage with 50% of xique-xique juice.

**Table 2 foods-10-01970-t002:** Physical and physicochemical parameters of formulated beverages during 21 days of refrigeration storage (4 °C).

Parameters	Time (Days)	Formulations			
		CB	B30	B40	B50
**pH**	1	3.14 ± 0.01 ^aA^	3.19 ± 0.00 ^aA^	3.19 ± 0.01 ^aA^	3.09 ± 0.01 ^aA^
	7	2.93 ± 0.01 ^bB^	3.00 ± 0.01 ^abA^	3.01 ± 0.01 ^aAB^	2.99 ± 0.01 ^abAB^
	14	2.86 ± 0.00 ^aB^	2.92 ± 0.02 ^aAB^	2.95 ± 0.01 ^aBC^	2.97 ± 0.01 ^aBC^
	21	2.80 ± 0.00 ^bC^	2.90 ± 0.01 ^abB^	2.93 ± 0.01 ^aC^	2.91 ± 0.01 ^abC^
**TA ^1^ (g/100 mL)**	1	11.66 ± 0.13 ^aB^	9.42 ± 0.08 ^bB^	8.66 ± 0.08 ^cB^	7.99 ± 0.13 ^dA^
	7	12.21 ± 0.26 ^aAB^	9.59 ± 0.07 ^bAB^	8.83 ± 0.07 ^cAB^	7.90 ± 0.08 ^dA^
	14	12.47 ± 0.27 ^aAB^	10.27 ± 0.13 ^bA^	9.21 ± 0.19 ^cAB^	8.66 ± 0.29 ^cA^
	21	12.76 ± 0.27 ^aA^	10.10 ± 0.27 ^bAB^	9.51 ± 0.13 ^bA^	9.51 ± 0.49 ^bA^
**TSS ^2^ (°Brix)**	1	6.90 ± 0.01 ^aB^	5.95 ± 0.05 ^bB^	6.10 ± 0.05 ^bB^	5.75 ± 0.05 ^cC^
	7	7.40 ± 0.05 ^aB^	6.95 ± 0.05 ^bA^	6.20 ± 0.10 ^dB^	6.65 ± 0.05 ^cB^
	14	7.85 ± 0.05 ^aA^	7.00 ± 0.10 ^bA^	6.80 ± 0.05 ^bA^	6.80 ± 0.05 ^bB^
	21	8.00 ± 0.05 ^aA^	7.00 ± 0.10 ^cA^	6.95 ± 0.05 ^cA^	7.50 ± 0.05 ^bA^
**Total ash (g/100 mL)**	1	0.26 ± 0.03 ^bA^	0.39 ± 0.01 ^aA^	0.40 ± 0.04 ^aAB^	0.47 ± 0.06 ^aAB^
	7	0.21 ± 0.01 ^bA^	0.35 ± 0.04 ^aA^	0.39 ± 0.02 ^aAB^	0.42 ± 0.01 ^aAB^
	14	0.19 ± 0.03 ^bA^	0.31 ± 0.05 ^abA^	0.40 ± 0.02 ^aA^	0.38 ± 0.05 ^aA^
	21	0.16 ± 0.01 ^bA^	0.26 ± 0.06 ^abA^	0.31 ± 0.01 ^aB^	0.30 ± 0.03 ^aB^
**Total sugars** **(g/100 mL)**	1	9.10 ± 0.10 ^aB^	7.97 ± 0.11 ^bD^	8.95 ± 0.10 ^aB^	8.31 ± 0.10 ^bB^
	7	10.24 ± 0.06 ^aA^	8.87 ± 0.03 ^bC^	9.07 ± 0.12 ^bB^	8.98 ± 0.26 ^bAB^
	14	10.68 ± 0.27 ^aA^	9.12 ± 0.07 ^bB^	10.00 ± 0.18 ^aA^	9.24 ± 0.05 ^bA^
	21	10.70 ± 0.29 ^aA^	9.64 ± 0.10 ^bA^	10.34 ± 0.20 ^aA^	9.80 ± 0.43 ^abA^
**Proteins (g/100 mL)**	1	0.58 ± 0.01 ^aA^	0.46 ± 0.01 ^bA^	0.46 ± 0.02 ^bcA^	0.41 ± 0.01 ^cA^
	7	0.55 ± 0.02 ^aA^	0.49 ± 0.04 ^aA^	0.50 ± 0.03 ^aA^	0.45 ± 0.05 ^aA^
	14	0.51 ± 0.02 ^aA^	0.46 ± 0.06 ^aA^	0.45 ± 0.05 ^aA^	0.45 ± 0.06 ^aA^
	21	0.56 ± 0.05 ^aA^	0.49 ± 0.04 ^aA^	0.47 ± 0.07 ^aA^	0.46 ± 0.05 ^aA^
**Lipids (g/100 mL)**	1	0.97 ± 0.14 ^aA^	0.67 ± 0.14 ^abA^	0.68 ± 0.18 ^abA^	0.42 ± 0.02 ^bA^
	7	0.91 ± 0.10 ^aA^	0.77 ± 0.33 ^aA^	0.68 ± 0.33 ^aA^	0.48 ± 0.37 ^aA^
	14	0.99 ± 0.07 ^aA^	0.71 ± 0.11 ^abA^	0.62 ± 0.10 ^bA^	0.48 ± 0.29 ^abA^
	21	1.00 ± 0.01 ^aA^	0.80 ± 0.27 ^aA^	0.70 ± 0.35 ^aA^	0.48 ± 0.37 ^aA^

Results are expressed as average (*n* = 3) ± standard deviation; ^1^ TA (Tritatable Acidity) is expressed as g citric acid per 100 mL of beverage; ^2^ TSS (Total Soluble Solids) is expressed as °Brix (25 °C); ^a–d^: mean ± standard deviation with different lowercase letters on the same storage time differ (*p* < 0.05) among beverage formulations, based on Tukey’s test; ^A–D^: mean ± standard deviation with different uppercase letters on the same beverage formulation differ (*p* < 0.05) among different storage time periods, based on Tukey’s test.

**Table 3 foods-10-01970-t003:** Soluble sugars and organic acids content of formulated beverages during 21 days of refrigeration storage (4 °C).

Parameters (g/100 mL)	Time (Days)	Beverages			
		CB	B30	B40	B50
**Soluble sugars**					
Glucose	1	1.00 ± 0.15 ^abA^	0.68 ± 0.07 ^abA^	0.55 ± 0.01 ^bB^	0.94 ± 0.01 ^aA^
	14	1.03 ± 0.21 ^abA^	0.83 ± 0.02 ^bA^	0.98 ± 0.09 ^abAB^	1.05 ± 0.01 ^aA^
	21	1.05 ± 0.01 ^aA^	1.02 ± 0.11 ^aA^	1.02 ± 0.04 ^aA^	1.09 ± 0.03 ^aA^
Fructose	1	0.87 ± 0.18 ^aA^	0.75 ± 0.07 ^aA^	0.67 ± 0.02 ^aB^	0.87 ± 0.01 ^aA^
	14	0.97 ± 0.04 ^aA^	0.79 ± 0.03 ^aA^	0.82 ± 0.08 ^aAB^	0.90 ± 0.01 ^aA^
	21	1.04 ± 0.15 ^aA^	0.85 ± 0.09 ^aA^	0.98 ± 0.04 ^aA^	1.03 ± 0.04 ^aA^
**Organic Acids**					
Citric	1	0.97 ± 0.14 ^aA^	0.83 ± 0.06 ^aA^	0.73 ± 0.05 ^aA^	0.72 ± 0.01 ^aA^
	14	1.01 ± 0.09 ^abA^	0.84 ± 0.06 ^abA^	0.82 ± 0.01 ^aA^	0.74 ± 0.01 ^bA^
	21	1.12 ± 0.03 ^aA^	0.89 ± 0.02 ^bA^	0.82 ± 0.02 ^bA^	0.76 ± 0.01 ^cA^
Malic	1	0.07 ± 0.02 ^aA^	0.07 ± 0.01 ^aA^	0.06 ± 0.01 ^aAB^	0.07 ± 0.00 ^aA^
	14	0.06 ± 0.01 ^aA^	0.06 ± 0.02 ^aA^	0.04 ± 0.00 ^aB^	0.03 ± 0.00 ^aB^
	21	0.09 ± 0.00 ^aA^	0.09 ± 0.01 ^aA^	0.09 ± 0.00 ^aA^	0.06 ± 0.00 ^aA^
Succinic	1	0.04 ± 0.01 ^aA^	0.03 ± 0.01 ^aA^	0.03 ± 0.00 ^aA^	0.02 ± 0.00 ^aA^
	14	0.01 ± 0.00 ^aA^	0.01 ± 0.00 ^aA^	0.01 ± 0.00 ^aA^	0.01 ± 0.00 ^aB^
	21	0.01 ± 0.00 ^aA^	0.01 ± 0.00 ^aA^	0.02 ± 0.00 ^aA^	0.01 ± 0.00 ^aAB^
Lactic	1	0.01 ± 0.00 ^aA^	0.02 ± 0.00 ^aA^	0.02 ± 0.00 ^aAB^	0.02 ± 0.00 ^aA^
	14	0.03 ± 0.02 ^aA^	0.03 ± 0.01 ^aA^	0.02 ± 0.00 ^aB^	0.02 ± 0.00 ^aA^
	21	0.02 ± 0.00 ^bA^	0.02 ± 0.00 ^bA^	0.03 ± 0.00 ^aA^	0.03 ± 0.00 ^aA^

Results are expressed as average (*n* = 3) ± standard deviation; ^a–c^: mean ± standard deviation with different lowercase letters on the same storage time differ (*p* < 0.05) among beverage formulations, based on Tukey’s test; ^A–B^: mean ± standard deviation with different uppercase letters on the same beverage formulation differ (*p* < 0.05) among different storage time periods, based on Tukey’s test.

**Table 4 foods-10-01970-t004:** Phenolic compounds (mean ± standard deviation) identified in formulated beverages during 21 days of refrigeration storage (4 °C).

Phenolic Compounds (mg/100 g)	Time (Days)	Beverages			
		CB	B30	B40	B50
**Flavanones**					
Hesperidin	1	0.08 ± 0.01 ^aA^	0.05 ± 0.00 ^aA^	<LOD	<LOD
	14	0.07 ± 0.00 ^aA^	0.05 ± 0.00 ^aA^	<LOD	<LOD
	21	0.07 ± 0.01 ^aA^	0.05 ± 0.00 ^aA^	<LOD	<LOD
Naringenin	1	0.02 ± 0.00 ^a^	0.02 ± 0.00 ^aA^	0.02 ± 0.00 ^a^	0.03 ± 0.00 ^a^
	14	<LOD	0.02 ± 0.00 ^A^	<LOD	<LOD
	21	<LOD	<LOD	<LOD	<LOD
**Flavonols**					
Kaempferol	1	0.13 ± 0.02 ^aA^	0.13 ± 0.00 ^aA^	0.13 ± 0.00 ^aA^	0.13 ± 0.00 ^aA^
	14	0.13 ± 0.00 ^aA^	0.12 ± 0.01 ^aA^	0.13 ± 0.02 ^aAB^	0.13 ± 0.00 ^aA^
	21	0.13 ± 0.01 ^aA^	0.11 ± 0.01 ^aA^	0.12 ± 0.01 ^aB^	0.12 ± 0.01 ^aA^
Myricetin	1	0.37 ± 0.05 ^aA^	0.35 ± 0.00 ^aA^	0.37 ± 0.06 ^aA^	0.36 ± 0.02 ^aA^
	14	0.34 ± 0.01 ^aA^	0.32 ± 0.02 ^aA^	0.35 ± 0.02 ^aA^	0.36 ± 0.00 ^aA^
	21	0.33 ± 0.02 ^aA^	0.30 ± 0.01 ^aA^	0.33 ± 0.01 ^aA^	0.35 ± 0.00 ^aA^
Quercitin	1	0.05 ± 0.00 ^aA^	0.04 ± 0.00 ^aA^	0.04 ± 0.00 ^aA^	0.04 ± 0.00 ^aA^
	14	0.05 ± 0.00 ^aA^	0.05 ± 0.01 ^aA^	0.04 ± 0.00 ^aA^	0.04 ± 0.00 ^aA^
	21	0.04 ± 0.01 ^aA^	0.04 ± 0.00 ^aA^	0.04 ± 0.00 ^aA^	0.04 ± 0.00 ^aA^
Rutin	1	0.03 ± 0.01 ^aA^	0.03 ± 0.02 ^aA^	0.04 ± 0.01 ^aA^	0.05 ± 0.00 ^aA^
	14	0.03 ± 0.00 ^aA^	0.03 ± 0.02 ^aA^	0.03 ± 0.00 ^aA^	0.03 ± 0.00 ^aAB^
	21	0.02 ± 0.01 ^aA^	0.02 ± 0.00 ^aA^	0.02 ± 0.00 ^aA^	0.02 ± 0.00 ^aB^
**Hydroxybenzoic acids**					
Syringic acid	1	0.03 ± 0.01 ^aA^	0.02 ± 0.00 ^aA^	0.02 ± 0.00 ^aA^	0.01 ± 0.00 ^aA^
	14	0.03 ± 0.00 ^aA^	0.02 ± 0.00 ^abA^	0.02 ± 0.00 ^abA^	0.01 ± 0.00 ^bA^
	21	0.02 ± 0.00 ^aA^	0.02 ± 0.00 ^abA^	0.01 ± 0.00 ^bA^	0.01 ± 0.00 ^bA^
**Hydroxynamic acids**					
Caffeic acid	1	0.02 ± 0.00 ^aA^	0.01 ± 0.00 ^aA^	0.01 ± 0.00 ^aA^	0.01 ± 0.00 ^aA^
	14	0.02 ± 0.00 ^aA^	0.01 ± 0.00 ^aA^	0.01 ± 0.00 ^aA^	0.01 ± 0.00 ^aA^
	21	0.01 ± 0.00 ^aA^	0.01 ± 0.00 ^aA^	0.01 ± 0.00 ^aA^	0.01 ± 0.00 ^aA^
Coumaric acid	1	0.02 ± 0.00 ^aA^	0.01 ± 0.00 ^aA^	0.01 ± 0.01 ^aA^	0.01 ± 0.00 ^aA^
	14	0.02 ± 0.00 ^aA^	0.01 ± 0.00 ^aA^	0.01 ± 0.00 ^aA^	0.01 ± 0.00 ^aA^
	21	0.02 ± 0.00 ^aA^	0.01 ± 0.00 ^aA^	0.01 ± 0.00 ^aA^	0.01 ± 0.00 ^aA^
Caftaric acid	1	0.05 ± 0.01 ^aA^	0.05 ± 0.00 ^aA^	0.05 ± 0.00 ^aA^	0.05 ± 0.00 ^aA^
	14	0.05 ± 0.00 ^aA^	0.05 ± 0.00 ^aA^	0.05 ± 0.02 ^aA^	0.05 ± 0.00 ^aA^
	21	0.03 ± 0.00 ^aA^	0.03 ± 0.00 ^aA^	0.04 ± 0.00 ^aA^	0.05 ± 0.00 ^aA^
Chlorogenic acid	1	0.07 ± 0.00 ^abA^	0.07 ± 0.00 ^aA^	0.05 ± 0.01 ^abA^	0.04 ± 0.00 ^bA^
	14	0.07 ± 0.01 ^aAB^	0.05 ± 0.00 ^aA^	0.04 ± 0.01 ^aA^	0.04 ± 0.00 ^aA^
	21	0.06 ± 0.00 ^aB^	0.05 ± 0.00 ^aA^	0.04 ± 0.00 ^abA^	0.04 ± 0.00 ^bA^
**Polyphenols**					
Trans-resveratrol	1	0.02 ± 0.00 ^aA^	0.02 ± 0.00 ^aAB^	0.02 ± 0.00 ^aA^	0.02 ± 0.00 ^aA^
	14	0.02 ± 0.00 ^aA^	0.02 ± 0.00 ^aA^	0.02 ± 0.00 ^aA^	0.02 ± 0.00 ^aA^
	21	0.02 ± 0.00 ^aA^	0.01 ± 0.00 ^aB^	0.02 ± 0.00 ^aA^	0.02 ± 0.00 ^aA^
Cis-resveratrol	1	0.02 ± 0.00 ^aA^	0.02 ± 0.00 ^aA^	0.02 ± 0.00 ^aA^	0.02 ± 0.00 ^aA^
	14	0.02 ± 0.00 ^aA^	0.02 ± 0.00 ^aA^	0.02 ± 0.00 ^aA^	0.02 ± 0.00 ^aA^
	21	0.02 ± 0.00 ^aA^	0.02 ± 0.00 ^aA^	0.02 ± 0.00 ^aA^	0.02 ± 0.00 ^aA^
Epicatechins gallate	1	0.17 ± 0.00 ^aA^	0.11 ± 0.00 ^abA^	0.10 ± 0.00 ^bA^	0.08 ± 0.00 ^cA^
	14	0.16 ± 0.02 ^aA^	0.10 ± 0.00 ^aB^	0.09 ± 0.00 ^aA^	0.06 ± 0.01 ^aAB^
	21	0.15 ± 0.01 ^aA^	0.09 ± 0.03 ^abAB^	0.07 ± 0.01 ^abA^	0.05 ± 0.00 ^bB^
Epicatechins	1	0.04 ± 0.01 ^aA^	0.02 ± 0.00 ^aA^	0.02 ± 0.00 ^aA^	0.02 ± 0.00 ^aA^
	14	0.04 ± 0.00 ^aA^	0.02 ± 0.00 ^abA^	0.02 ± 0.00 ^bA^	0.02 ± 0.00 ^bB^
	21	0.03 ± 0.00 ^aA^	0.02 ± 0.00 ^aA^	0.01 ± 0.00 ^bA^	0.01 ± 0.00 ^bB^
**Flavanols**					
Epigallocatechin gallate	1	0.25 ± 0.05 ^aA^	0.16 ± 0.00 ^abA^	0.15 ± 0.03 ^abA^	0.12 ± 0.00 ^bA^
	14	0.23 ± 0.01 ^aAB^	0.14 ± 0.01 ^bAB^	0.14 ± 0.01 ^bA^	0.11 ± 0.01 ^bA^
	21	0.18 ± 0.01 ^aB^	0.11 ± 0.01 ^abB^	0.12 ± 0.03 ^abA^	0.09 ± 0.02 ^bA^
**Anthocyanins**					
Procyanidin B1	1	0.06 ± 0.01 ^aA^	0.05 ± 0.00 ^aA^	0.06 ± 0.01 ^aA^	0.05 ± 0.00 ^aA^
	14	0.05 ± 0.01 ^aA^	0.04 ± 0.00 ^aA^	0.05 ± 0.00 ^aA^	0.05 ± 0.00 ^aA^
	21	0.05 ± 0.00 ^aA^	0.04 ± 0.00 ^aA^	0.05 ± 0.00 ^aA^	0.05 ± 0.00 ^aA^
Procyanidin A2	1	0.13 ± 0.00 ^cA^	0.14 ± 0.00 ^bA^	0.14 ± 0.00 ^bA^	0.15 ± 0.00 ^aA^
	14	0.13 ± 0.00 ^bAB^	0.13 ± 0.01 ^abA^	0.14 ± 0.2 ^abAB^	0.14 ± 0.00 ^aB^
	21	0.12 ± 0.01 ^bB^	0.12 ± 0.00 ^bB^	0.13 ± 0.00 ^aB^	0.13 ± 0.00 ^aC^

Results are expressed as average (*n* = 3) ± standard deviation; ^a–c^: mean ± standard deviation with different lowercase letters on the same storage time differ (*p* < 0.05) among beverage formulations, based on Tukey’s test; ^A–C^: mean ± standard deviation with different uppercase letters on the same beverage formulation differ (*p* < 0.05) among different storage time periods, based on Tukey’s test; < LOD: below the limit of detection.

## Data Availability

All data generated or analyzed during this study are included in this published article.

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
