# Peer review of "Physicochemical Parameters, Phytochemical Profile and Antioxidant Properties of a New Beverage Formulated with Xique-Xique (Pilosocereus gounellei) Cladode Juice"

_foods, 2021, doi:10.3390/foods10091970_

Round 1
Reviewer 1 Report
REVIEW
LINE 107-122 The preservation method is not the best that could be used. This is because you employed very high temperature and just refrigeration instead of freezing at less than -20 oC before used or analysed. Ion adition cotton mesh is not the right filter for juice
LINES 124-125. The recipes the authors used are not decided by a statistical plan and therefore the results are not expected to approach an optimum.
LINE 221 Why FRAP and ABTS and not DPPH???? Is there any specific reason for it?
Lines 264-267 “The increasing acidity found during the measured refrigerated storage period could 264 be caused by degradation of polyphenols in beverage formulations over time [9]. This 265 gradual increase in acidity could be a positive feature in formulated beverages, as a low 266 pH could inhibit the growth of pathogenic and spoilage microorganisms”
I strongly disagree with the above explanation I believe that the reason is microbiological action on the sugars and not on the polyphenols. In addition there is another problematic and hazardous point. For pH over 4.6. there is always the possibility of poisoning due to Clostridium Botulinum.
It obvious according to my opinion that the experimental work has got serious and problematic points and it is not well organized. There is not any statistical plan to select the mixing ratios of the constituents of the juice and in addition the preservation of a juice in cool storage and not in rerigaration is immposible evan if it is abused with heating at 100 C as it happened in this work.
Author Response
Dear,
We are sending to you the revised version of the manuscript ID: foods-1314376, entitled “Physicochemical parameters, phytochemical profile and antioxidant properties of a new beverage formulated with xique-xique (Pilosocereus gounellei) cladode and passion fruit (Passiflora edulis) juice” in consonance with reviewer’ suggestions. Reviewer’ comments are in “black” and our answers are in “red”. Changes in the revised version of the manuscript were marked using the “Track Changes” function.
We thank the Editor and the reviewer for the opportunity to review and improve our work to be reconsidered. Several parts of the manuscript where the reviewers pointed out the need for improvement, as well as parts that we think are important to change, have been rewritten, as can be seen in file labeled 'Revised Manuscript with Track Changes'.
All suggestions were considered to improve the general quality of the revised manuscript and to fit its format to the requirement of Foods.
Reviewer #1:
Point 1: LINE 107-122 The preservation method is not the best that could be used. This is because you employed very high temperature and just refrigeration instead of freezing at less than -20 °C before used or analyzed. In addition, cotton mesh is not the right filter for juice.
Response 1: One of the intention of this study was to evaluate the impact of refrigeration storage on bioactive compounds found in the formulated beverages, simulating how they would be stored if were commercially distributed, which is why we have chosen refrigeration temperature rather than freezing. Regarding the preservation method, we would like to rectify that the duration was 5 minutes instead of 10 minutes. This correction was also made in the manuscript on page 3, line 114.
Regarding this aspect of the processing of xique-xique juice, we reiterate that after heat treatment (90 °C, 5 min), it was cooled to 10 ± 1 °C and then stored at refrigeration temperature (4 ± 1 °C) until the moment of processing of the beverages added of this juice.
As for the high temperature applied, other recent studies evaluating bioactive compounds in vegetable beverages, published previously in Foods, have also used similar temperature and time as used in our study, which are cited below:
Bianchi, F.; Punsch, M.; Venir, E. Effect of processing and storage on the quality of beetroot and apple mixed juice. Foods 2021, 10, 1052, doi:10.3390/foods10051052
Škegro, M.; Putnik, P.; Kovačević, D.B.; Kovač, A.P.; Salkić, L.; Čanak, I.; … & Ježek, D. Chemometric comparison of high-pressure processing and thermal pasteurization: the nutritive, sensory, and microbial quality of smoothies. Foods 2021, 10, 1167, doi:10.3390/foods10061167
As for the filtration method, we realize that “cotton mesh” is not the most adequate term for the material we used. “Muslin cloth” is the correct term, which is an appropriate filter for vegetable and fruit juices, as shown in published studies that can be consulted below. See in page 3, line 114 the changes made in the revised manuscript.
Ani, P.N.; Abel, H.C. Nutrient, phytochemical. and antinutrient composition of Citrus maxima fruit juice and peel extract. Food Sci. Nutr. 2018, 6, 653-658, doi:10.1002/fsn3.604
Baccouche, A.; Ennouri, M.; Felfould, I.; Attia, H. A physical stability study of whey-based prickly pear beverage. Food Hydrocoll. 2013, 33, 234-244, doi:10.1016/j.foodhyd.2013.03.007
Lakshan, S.A.T.; Jayanath, N.Y.; Abeysekera, W.P.K.M.; Abeysekera, W.K.S.M. A commercial potential blue pea (Clitoria ternatea L.) flower extract incorporated beverage having functional properties. Evid. Based Complement. Alternat. Med. 2019, 2019, 1-13, doi:10.1155/2019/2916914
Mishra, L.K.; Sangma, D. Quality attributes, phytochemical profile and storage stability studies of functional ready to serve (RTS) drink made from blend of Aloe vera, sweet lime, amla and ginger. J. Food Sci. Technol. 2017, 54, 761-769, doi:10.1007/s13197-017-2516-9.
Point 2: LINES 124-125. The recipes the authors used are not decided by a statistical plan and therefore the results are not expected to approach an optimum.
Response 2: The recipes were decided in a pilot study based on sensory attributes and in previous studies with functional mixed beverages development, which can be consulted below. It is worth of pointing out that the recipes in these published studies were also not decided by a statistical plan.
Gironés-Vilaplana, A.; Mena, P.; García-Viguera, C.; Moreno, D.A. A novel beverage rich in antioxidant phenolics: maqui berry (Aristotelia chilensis) and lemon juice. LWT 2012, 47, 279-286, doi:10.1016/j.lwt.2012.01.020.
Mishra, L.K.; Sangma, D. Quality attributes, phytochemical profile and storage stability studies of functional ready to serve (RTS) drink made from blend of Aloe vera, sweet lime, amla and ginger. J. Food Sci. Technol. 2017, 54, 761-769, doi:10.1007/s13197-017-2516-9.
Beverage’s formulations were also decided based on xique-xique cladodes juice composition, which was evaluated in previous studies:
Ribeiro, T.S.; Sampaio, K.B.; Menezes, F.N.D.D.; Assis, P.O.A.; Lima, M.S.; Oliveira, M.E.G.; Souza, E.L.; Queiroga, R. C. R. E. In vitro evaluation of potential prebiotic effects of a freeze‑dried juice from Pilosocereus gounellei (A. Weber ex K. Schum. Bly. Ex Rowl) cladodes, an unconventional edible plant from Caatinga biome. 3 Biotech 2020, 10, 1-9. doi:10.1007/s13205-020-02442-8.
Carvalho, P.O.A.A.; Guerra, G.C.B.; Borges, G.S.C.; Bezerril, F.F.; Sampaio, K.B.; Ribeiro, T.S.; Pacheco, M.T.B.; Milani, R.F.; Goldbeck, R.; Ávila, P.F. Nutritional potential and bioactive compounds of xique-xique juice: an unconventional food plant from Semiarid Brazilian. J. Food Process. Preserv. 2021, 45, 1-10, doi:10.1111/jfpp.15265.
Point 3: LINE 221 Why FRAP and ABTS and not DPPH???? Is there any specific reason for it?
Response 3: Due to the fact that reactions involving antioxidant activity are complex, these types of assays should not be evaluated by a single method. For this reason, in the present study, the antioxidant activity of beverages was evaluated using the FRAP and ABTS methods. The FRAP method has the principle of reducing the ferric-tripyridyltriazine complex to a ferrous complex in the presence of an antioxidant under acidic conditions. Due to the fact that the beverage examined in the present study had an acidic pH (range 2.80-3.19), the FRAP method was chosen.
The ABTS and DPPH assays are methods whose action mechanism is the capture of free radicals. Both methods have good stability, differing in terms of handling (DPPH is a free radical that is acquired in this way, without the need for preparation; the ABTS radical must be generated by enzymatic or chemical reactions). Considering that the methods have similar principles, some characteristics, such as solubilization in aqueous and organic media (ABTS) and not specifically in organic media (DPPH) had directed us to choose the ABTS method.
Furthermore, both ABTS and FRAP assays are methods with easy laboratory reproducibility and widely used for the determination of the antioxidant activity in fruits and beverages, with numerous publications, as can be seen in studies cited below:
Castro, J.M.C.; Alves, C.A.N.; Santos, K.L.; Silva, E.O.; Araújo, I.M.S.; Vasconcelos, L.B. Elaboration of a mixed beverage from hibiscus and coconut water: an evaluation of bioactive and sensory properties. Int. J. Gastron. Food Sci. 2021, 23, 100284,1-100284:8, doi:10.1016/j.ijgfs.2020.100284.
Gironés-Vilaplana, A.; Mena, P.; García-Viguera, C.; Moreno, D.A. A novel beverage rich in antioxidant phenolics: maqui berry (Aristotelia chilensis) and lemon juice. LWT 2012, 47, 279-286, doi:10.1016/j.lwt.2012.01.020.
Reis, L.C.R.; Facco, E.M.P.; Flôres, S.H.; Rios, A.O. Stability of functional compounds and antioxidant activity of fresh and pasteurized orange passion fruit (Passiflora caerulea) during cold storage. Food Res. Int. 2018, 106, 481-486, doi:10.1016/j.foodres.2018.01.019.
Wurlitzer, N.J.; Dionísio, A.P.; Lima, J.R.; Garruti, D.S.; Araújo, I.M.S.; Rocha, R.F.J.; Maia, J.L. Tropical fruit juice: effect of thermal treatment and storage time on sensory and functional properties. J. Food Sci. Technol. 2019, 56, 5184-5193, doi:10.1007/s13197-019-03987-0.
Point 4: Lines 264-267 “The increasing acidity found during the measured refrigerated storage period could be caused by degradation of polyphenols in beverage formulations over time [9]. This gradual increase in acidity could be a positive feature in formulated beverages, as a low pH could inhibit the growth of pathogenic and spoilage microorganisms”. I strongly disagree with the above explanation I believe that the reason is microbiological action on the sugars and not on the polyphenols. In addition there is another problematic and hazardous point. For pH over 4.6. there is always the possibility of poisoning due to Clostridium botulinum.
Response 4: Although there was an increase in acidity, no significant decrease in total sugar contents, TSS, glucose or fructose was observed in our study through the measured storage time, disproving a possible sugar degradation by microorganisms. However, a significant decrease in total phenolic compounds can be observed through the storage time, as shown in Figure 1A, confirming their degradation. This same acidity increase behavior correlated with the degradation of phenolic compounds was reported by other studies, as cited below:
Mishra, L.K.; Sangma, D. Quality attributes, phytochemical profile and storage stability studies of functional ready to serve (RTS) drink made from blend of Aloe vera, sweet lime, amla and ginger. J. Food Sci. Technol. 2017, 54, 761-769, doi:10.1007/s13197-017-2516-9. doi:10.1007/s13197-017-2516-9.
Nidhi, G.R., Singh, R., Rana, M.K. Changes in chemical composition of ready-to-serve bael-guava blended beverage during storage. J. Food Sci Technol. 2008, 45, 378–380.
Ramachandran, P., Nagarajan, S. Quality characteristics, nutraceutical profile and storage stability of the aloe gel-papaya functional beverage blend. Int. J. Food Sci. 2014, 1-7, doi:10.1155/2014/847013
Yadav, R., Tripathi, A.D., Jha, A. Effect of storage time on the physicochemical properties and sensory attributes of Aloe vera ready-to-serve (RTS) beverage. Int. J. Food Nutr. Public Health 2013, 6, 172–193.
As for the possibility of C. botulinum poisoning, no examined beverage had pH over 4.6 in this study. Maximum pH value found was of 3.19.
Point 5: It obvious according to my opinion that the experimental work has got serious and problematic points and it is not well organized. There is not any statistical plan to select the mixing ratios of the constituents of the juice and in addition the preservation of a juice in cool storage and not in rerigaration is immposible evan if it is abused with heating at 100 °C as it happened in this work.
Response 4: We appreciate your comments and observations. Regarding this aspect of the processing of xique-xique juice, we reiterate that after heat treatment (90 °C, 5 min), it was cooled to 10 ± 1 °C and then stored at refrigeration temperature (4 ± 1 °C) until the moment of processing the beverages added of this juice. Minor changes were made in page 3, line 115 to clarify this information. The other points have already been clarified previously.

Reviewer 2 Report
The topic of the submitted manuscript, entitled: “Physicochemical parameters, phytochemical profile and antioxidant properties of a new beverage formulated with xique xique (Pilosocereus gounellei) cladode and passion fruit (Passiflora edulis) juice”. The introduction is well written, and the methods are clearly described. This work has a potential interest and fits with the scope of the journal. However, authors should make small changes to improve its quality. Consequently, I think minor revision work is needed.
Detailed comments:
In my opinion, the title of the manuscript should be shorten.
Sensory acceptance is very important in food evaluation. What was the sensory quality of these juices?
Methods
L.147: “Lipid content was measured by cold extraction method [27].” - Details of the analysis are needed because it describes the method of extracting lipids from animal tissues.
All abbreviations should be explained in this manuscript., e.g. L.259: “TA”
L.260: Control formulation (BC) but L.262 is CB, which is correct?
L261-263: “This variation could be attributed to the higher concentration of passion fruit juice in CB, since it is a more acidic juice with pH of approximately 2.8, while xique-xique cladode juice has a pH of approximately 5.0 [35].” Is it the authors' data or information from the literature?
Regarding Table3 and Table 4.
Please do explain: why the samples were not, tested after 7 days of storage?.
Checking the statistical analysis in Table 4 is needed. For example: “Trans-resveratrol” (after 1 day storage) is:
|
0.02 ±0.00abA |
0.02 ±0.00aAB |
0.02 ±0.00abAB |
0.02 ±0.00bA ? |
L.476- 480: “Regarding correlations among antioxidant assays and bioactive compounds…” - It is not clear where this results are located?
Author Response
Dear,
We are sending to you the revised version of the manuscript ID: foods-1314376, entitled “Physicochemical parameters, phytochemical profile and antioxidant properties of a new beverage formulated with xique-xique (Pilosocereus gounellei) cladode and passion fruit (Passiflora edulis) juice” in consonance with reviewer’ suggestions. Reviewer’ comments are in “black” and our answers are in “red”. Changes in the revised version of the manuscript were marked using the “Track Changes” function.
We thank the Editor and the reviewer for the opportunity to review and improve our work to be reconsidered. Several parts of the manuscript where the reviewers pointed out the need for improvement, as well as parts that we think are important to change, have been rewritten, as can be seen in file labeled 'Revised Manuscript with Track Changes'.
All suggestions were considered to improve the general quality of the revised manuscript and to fit its format to the requirement of Foods.
Reviewer #2:
The topic of the submitted manuscript, entitled: “Physicochemical parameters, phytochemical profile and antioxidant properties of a new beverage formulated with xique xique (Pilosocereus gounellei) cladode and passion fruit (Passiflora edulis) juice”. The introduction is well written, and the methods are clearly described. This work has a potential interest and fits with the scope of the journal. However, authors should make small changes to improve its quality. Consequently, I think minor revision work is needed.
Point 1: In my opinion, the title of the manuscript should be shorten.
Response 1: We appreciate your suggestion. See changes made on page 1, lines 2-5: “Physicochemical parameters, phytochemical profile and antioxidant properties of a new beverage formulated with xique xique (Pilosocereus gounellei) cladode juice”.
Point 2: Sensory acceptance is very important in food evaluation. What was the sensory quality of these juices?
Response 2: We agreed with this reflection and sensory acceptance was an essay that we intended to carry out. However, the COVID-19 pandemic, still in progress, made it impossible for us to carry out this trial, in view of the recommendations of social isolation by the National and International Health Agencies. However, from pilot tests to standardize the formulations of elaborated beverages, we verified excellent sensory qualities, especially regarding the attributes of appearance, color, flavor, odor, consistency and global acceptance.
Point 3: L.147: “Lipid content was measured by cold extraction method [27].” - Details of the analysis are needed because it describes the method of extracting lipids from animal tissues.
Response 3: The requested details were added in lines 147-154 of the revised manuscript. Although the original method describes the lipid extraction from animal tissues, this method can also be applied in vegetable matrices, as shown in studies cited below:
Albuquerque, J.G., Escalona-Buendía, H.B., Cordeiro, A.M.T.M., Lima, M.S., Aquino, J.S., Vasconcelos, M.A.S. Ultrasound treatment for improving the bioactive compounds and quality properties of a Brazilian nopal (Opuntia ficus-indica) beverage during shelf-life. LWT 2021, 149,111814, doi:10.1016/j.lwt.2021.111814.
Pande, G.; Akoh, C.C. Organic acids, antioxidant capacity, phenolic content and lipid characterisation of Georgia-grown underutilized fruit crops. Food Chem. 2010, 120, 1067-1075, doi:10.1016/j.foodchem.2009.11.054.
Point 4: All abbreviations should be explained in this manuscript., e.g. L.259: “TA”
Response 4: The full term for the abbreviation was added in page 6, line 266, of the revised manuscript.
Point 5: L.260: Control formulation (BC) but L.262 is CB, which is correct?
Response 5: The correct abbreviation is CB (control beverage). The changes were made in lines 267 and 305 of the revised manuscript.
Point 6: L261-263: “This variation could be attributed to the higher concentration of passion fruit juice in CB, since it is a more acidic juice with pH of approximately 2.8, while xique-xique cladode juice has a pH of approximately 5.0 [35].” Is it the authors' data or information from the literature?
Response 6: Data regarding the pH of passion fruit juice and xique-xique cladode juice were taken from the literature. More information was added in lines 270-271 of the revised manuscript in order to clarify it.
Point 7: Regarding Table3 and Table 4. Please do explain: why the samples were not, tested after 7 days of storage?
Response 7: Previous studies carried out by our research group had demonstrated that for the variables presented in Tables 3 and 4 the results found on day 7 of storage did not differ significantly from those found on days 1 and 14 of storage. Therefore, we had chosen to bring the results from the beginning (time 1), middle (time 14) and end (time 21) of the measured storage period.
Point 8: Checking the statistical analysis in Table 4 is needed. For example: “Trans-resveratrol” (after 1 day storage) is:
|
0.02 ±0.00abA |
0.02 ±0.00aAB |
0.02 ±0.00abAB |
0.02 ±0.00bA ? |
Response 8: The statistical analysis was checked and corrections were made in Table 4.
Point 9: L.476- 480: “Regarding correlations among antioxidant assays and bioactive compounds…” - It is not clear where this results are located?
Response 9: These results are shown in Figure 3. We highlighted the location in the text. See page 14, line 486 of the revised manuscript. As for the “r” values, they have only been described in the text, as this variable is usually presented in this way, as can be seen in the studies cited bellow:
Andrés, V.; Tenorio, M.D.; Villanueva, M.J. Sensory profile, soluble sugars, organic acids, and mineral content in milk and soy-juice based beverages. Food Chem. 2015, 173, 1100-1106, doi:10.1016/j.foodchem.2014.10.136.
Dantas, A.M., Mafaldo, I.M., Oliveira, P.M.L., Lima, M.D.S., Magnani, M., Borges, G.D.S.C. Bioaccessibility of phenolic compounds in native and exotic frozen pulps explored in Brazil using a digestion model coupled with a simulated intestinal barrier. Food Chem. 2019, 15,202-214, doi:10.1016/j.foodchem.2018.08.099.
de Morais, J.S., Sant'Ana, A.S., Dantas, A.M., Silva, B.S., Lima, MS., Borges, G.C., Magnani, M. Antioxidant activity and bioaccessibility of phenolic compounds in white, red, blue, purple, yellow and orange edible flowers through a simulated intestinal barrier. Food Res. Int. 2020, 131,109046, doi:10.1016/j.foodres.2020.109046.
Oliveira, S.D.; Araújo, C.M.; Borges, G.S.C.; Lima, M.S.; Viera, V.B.; Garcia, E.F.; Souza, E.L.; Oliveira, M.E.G. Improvement in physicochemical characteristics, bioactive compounds and antioxidant activity of acerola (Malpighia emarginata D.C.) and guava (Psidium guajava L.) fruit by-products fermented with potentially probiotic lactobacilli. LWT 2020, 134, 110200, doi:10.1016/j.lwt.2020.110200.
